# The Development of Microscopy for Super-Resolution: Confocal Microscopy, and Image Scanning Microscopy

**Colin J. R. Sheppard** [1,2]

1    Nanophysics Istituto Italiano di Tecnologia, Via Enrico Melen, 83 Edificio B, 16152 Genova, Italy; colinjrsheppard@gmail.com

2    Molecular Horizons, School of Chemistry and Molecular Biosciences, University of Wollongong, Wollongong, NSW 2522, Australia

**Abstract:** Optical methods of super-resolution microscopy, such as confocal microscopy, structured illumination, nonlinear microscopy, and image scanning microscopy are reviewed. These methods avoid strong invasive interaction with a sample, allowing the observation of delicate biological samples. The meaning of resolution and the basic principles and different approaches to superresolution are discussed.

**Keywords:** super-resolution; confocal microscopy; image scanning microscopy; nonlinear microscopy; deconvolution

## 1. Introduction

Optical microscopy is a key instrumental technique in many areas of science and medicine. It is also widely used in an industrial setting and in mechanical, electrical, and chemical areas, especially in fault diagnosis and metrology. The main advantage of optical microscopy is its noninvasive nature, as compared with other, inherently invasive, types of microscopy, such as electron microscopy. However, optical microscopy is limited in its spatial resolution, classically at the level of about half a wavelength of the illuminating light. Recently, great progress has been made in overcoming this classical limit. In particular, approaches sometimes attributed to switching, such as stimulated emission depletion microscopy (STED) and localisation microscopy, have achieved outstanding resolution. However, to some degree these methods negate the main noninvasive advantage of optical microscopy by using intense illumination or chemical interaction. Hence, there is still strong interest in exploring improvements that can be achieved using more conventional optical techniques. Historically, the first successful approach was confocal microscopy. However, as much of the available light is rejected, the resolution must be traded off to attain an acceptable signal-to-noise ratio. Recently, several techniques have been introduced that overcome this compromise. The main aim of this paper is to bring out the fundamental unifying features of these methods. We start with a discussion of the nature of super-resolution and identify various different strategies, including optical and digital methods. We introduce the two alternative geometries for an imaging system. Then, we review confocal microscopy and discuss how the two imaging geoemetries can be combined. This leads to the concept of a confocal microscope with a detector array, or, equivalently, to a conventional microscope with patterned illumination. We are of the opinion that this general approach, combined with appropriate digital processing, can lead to further substantial improvement in the future.

## 2. Literature Review

### 2.1. Super-Resolution

2.1.1. Resolution Criteria and Resolution Limits

Before we can discuss super-resolution, we must consider what we mean by resolution. There are many different alternative resolution criteria. The simplest is based on the width of the central lobe of the point spread function (PSF), often defined as the full-width at half maximum (FWHM). While this can give a reasonable estimate in many cases, it is not a reliable measure. The two most well-known criteria are those due to Rayleigh and to Abbe.

Rayleigh's criterion is based on an object consisting of two self-luminous (i.e., incoherent) points of equal strength [1]. He claimed that the image of the points is resolved if the points are separated by a distance $1.22\lambda/NA$, where $\lambda$ is the wavelength of the light and $NA = n\sin\alpha$ is the numerical aperture, with $n$ the refractive index of the immersion medium and $\alpha$ the angle subtended by the edge of the lens pupil with respect to the optical axis. Rayleigh himself explained that this criterion is arbitrary and not a hard limit to resolution. Rayleigh's criterion was proposed for the special case of a circular aperture with no aberrations in an incoherent imaging system. As the intensity midway between the points when they are just resolved in this case is 0.735 times the intensity at the points, the criterion has been generalised by taking the points to be just resolved when this same ratio is achieved in the general case. The FWHM of the PSF is often consistent with the Rayleigh criterion, but not always. For example, pupil filters can sharpen the PSF but at the expense of an increased side-lobe level, which can worsen the Rayleigh two-point resolution. Additionally, coherent and incoherent systems have the same intensity PSF, but the Rayleigh resolution is better for the incoherent case.

Abbe's resolution limit is a well-defined transition from being resolved to being not resolved [2]. It is based on the image of a grating of lines. Abbe assumed the object to be illuminated by a plane wave. The grating diffracts light into different orders, which are either transmitted through the optical system or not. The angles of propagation of these orders are given by a Fourier series expansion of the grating structure, so there is a well-defined cut-off spatial frequency. In order to get an image of the grating, two diffracted orders must travel through the system. The cut-off frequency if the object is illuminated along the optical axis is $NA/\lambda$. Abbe was aware that illuminating at an oblique angle increases the cut-off frequency. If the illumination angle coincides with the edge of the aperture of the objective lens, then the cut-off is $2NA/\lambda$. However, Abbe also recognised that an image depends on the strength of the spatial frequency response in addition to the cut-off frequency, which is now described by a transfer function. For a coherent imaging system, we use a coherent transfer function (CTF), which operates on amplitudes, to describe the imaging properties. For incoherent imaging, we use an optical transfer function (OTF), which operates on intensities. A major limitation of Abbe's resolution limit is that it is based on coherent illumination, whereas in practice, microscopes used illumination with an incoherent source. Later, Rayleigh summed the illumination over a range of incident angles and showed that the result for the cut-off frequency can be up to $2NA/\lambda$ and that this is also the cut-off for a self-luminous (incoherent) object [3].

A more complete understanding of the effect of condenser aperture was achieved after the development of the theory of partial coherence [4]. Instead of a CTF or OTF, we have a transmission cross-coefficient (TCC) (sometimes called a partially-coherent transfer function) that describes the transfer of pairs of spatial frequencies into the intensity image. Different diffraction orders interfere and generate sum and difference frequencies in the image. We must now distinguish between the maximum spatial frequency of the object that is imaged, and the maximum spatial frequency in the intensity image [5].

Rayleigh also considered an object consisting of a bright feature, e.g., a line, on a bright background [3]. He found that there is no diffraction limit to the resolution in this case, but the quality of the image is just a question of contrast. This is important, as many papers have used measurements of similar structures to claim super-resolution. Rayleigh did not

show any plots of images, but these have been presented in Ref. [6]. This paper also gives a more detailed review of the subject of resolution.

### 2.1.2. Information Theory

The fact that we can do better than the resolution limit suggests that resolution is not an inherent property of an optical system. Instead, the optical system has the capacity to transmit a finite amount of information. The concept of information capacity was introduced by Hartley [7] and the concept of channel capacity in the presence of noise by Shannon [8]. Other important papers were by Gabor and Toraldo di Francia [9,10]. Lukosz stated that it is not the spatial frequency bandwidth but the number of degrees of freedom that can be transmitted by the system that is invariant and gave an expression for the information capacity of a two-dimensional (2D) imaging system, including time [11]. He explained that resolution can be increased by making some trade-off between the parameters, which requires that some a priori information concerning the object is known. Cox included depth information, time, and also the effect of noise in a 3D system [12]. This expression was later updated to include full 3D imaging, including polarisation [13], to give the number of degrees of freedom $N$ as

$$N = 3(8VB_v + 1)\log_2\left(1 + \frac{s}{n}\right), \tag{1}$$

where $s$ and $n$ are the signal and noise, respectively, $V$ is the volume of the object, and $B_v$ is the volume of the 3D transfer function. The theorem of invariance of the information capacity is the fundamental principle underlying all super-resolution schemes. We require sufficient information to allow the super-resolved image to be generated.

Cox investigated some examples of imaging systems [12]. Analytic continuation relies on the fact that the spatial frequency distribution of a spatially-limited object is analytic. Effectively, analytic continuation gives an increased spatial frequency bandwidth by trading off the signal-to-noise ratio (SNR). The gain in bandwidth, however, is very small for a substantial reduction in SNR. Further, if the object's field-of-view is limited to ∼20 resolution elements, a small additional improvement can be obtained. The incorporation of a priori information can also give further improvements.

### 2.1.3. Classes of Super-Resolution

Three distinct levels, or classes, of super-resolution have been identified [14,15].

1.  Ultra-resolution: Improved spatial frequency response and two-point resolution, but the cut-off is unchanged;
2.  Restricted super-resolution: the cut-off is increased, but less than $2n/\lambda$ (coherent) or $4n/\lambda$ (incoherent);
3.  Unrestricted super-resolution: the cut-off is increased without a limit.

The first class is not really super-resolution, as the spatial frequency cut-off is not increased, hence the different name: ultra-resolution. This class includes the use of pupil filters to sharpen the PSF. Toraldo di Francia showed that the width of the PSF can be made arbitrarily narrower by using a pupil mask, but with increased strength of sidelobes. The spatial frequency cut-off is, however, unchanged [16,17]. The effect is now sometimes described as super-oscillations. Other examples of this class include simple linear digital deconvolution, for example, based on Wiener filtering or the nearest neighbour method.

The next class is basically those systems where the band-width is doubled. This is achieved by modulation/demodulation (coding/decoding) and multiplexing. These systems combine patterned illumination with demodulation, which doubles the cut-off, and include the structured illumination microscope (SIM) and the confocal microscope. As the apertures of the illuminating system and the detection system are limited to $\sin \alpha = 1$, then for each of illumination and detection, $NA = n$, giving a maximum cut-off frequency of $2n/\lambda$. Here, we have assumed that the wavelengths of illumination and detection are equal, but if not, the cut-off frequency is the sum of the illumination and detection frequencies.

In the third class, the increase in the bandwidth is unlimited. These methods include STED [18], localisation microscopy [19–21], and fluorescence saturation [22] or switching, which, although they are important and give impressive resolution, are beyond the scope of this review. Other examples of this class use evanescent waves, including near-field microscopy, where a scanned, subwavelength aperture is used to illuminate the object. The first proposal of near-field microscopy was by Synge in 1928 [23]. Then the idea was reinvented in 1956 [24,25]. Baez demonstrated the method using acoustic waves, and Ash and Nicholls used microwaves in 1972 [26]. Then three groups all published optical demonstrations in 1984 [27–29]. Interestingly, this followed very soon after the invention of the scanning tunneling microscope in 1982, which stimulated the instrumental designs. Another approach, the photon-tunneling microscope, uses evanescent waves generated at an interface [30,31]. The aperture was avoided in apertureless near-field microscopy [32,33]. Another more recent example using evanescent waves is the perfect lens, using negative refractive index materials. Many methods of super-resolution rely on some form of nonlinearity, including digital deconvolution with constraints and nonlinear microscopy based on multiphoton fluorescence [34–36] or second harmonic generation (SHG) [37]. Furthermore, included in this class is pupil filters (super-oscillations) combined with a restricted field-of-view. In this case, the sidelobes are pushed outside of the field-of-view, and the spatial frequency cut-off increased. However, the pupil filter alone does not give an increased cut-off and is in the first class.

### 2.2. Imaging Geometries

### 2.2.1. Conventional Imaging

There are two ways to produce an image. The first is the direct, or conventional, approach. Light from the sample is imaged directly by a lens. This is, of course, the principle of vision in nature.

### 2.2.2. Scanned Imaging

The second way to form an image is by scanning. Bain filed a patent for using a scanning method to transmit an image using facsimile [38]. Later, scanning was used in the early development of television. A scanning optical microscope was developed by Roberts and Young [39]. In this system, which they called a flying-spot microscope, the object is illuminated with a scanned, focused spot of light. A signal is picked up with a single, bucket detector, and an image is built up by scanning in synchronism with the illuminating spot. Recently, such a system has also been called a single-pixel camera. The light emanating from the sample is then merely detected, rather than imaged. Therefore, unlike a conventional imaging system, the image is basically formed by the illumination system. Further, as the light from the sample is only detected, the direct line of sight from the object to the detector is unnecessary and can be obstructed by a scattering medium.

The principle of the flying-spot microscope was discussed earlier and attributed to Richard.C. Webb, by Zvorykin and Ramberg in 1949 [40], p.383. Interestingly, the flying-spot microscope was widely investigated in those days, culminating in a meeting of the New York Academy of Sciences in 1962. For some reason, interest in the topic declined soon after.

Scanning systems can also be used in various imaging modes where the illuminating light is used to produce some physical response that can be used to generate an image [41]. Examples include many types of spectroscopy, photoacoustic and photothermal generation, and the generation of charge carriers in semiconductors.

Using the principle of reciprocity, it can be shown that scanning and conventional systems usually give identical images, even in the presence of multiple scattering [42]. However, this property breaks down if there is a change in wavelength in the interaction with the sample, as in fluorescence with a Stokes shift or in two-photon microscopy. In the case of fluorescence, if a Stokes shift is present, then the scanning system gives a better resolution, as the wavelength of the illuminating light is shorter than the fluorescent light.

In the case of two-photon fluorescence (2PF) or SHG, the emitted radiation has a shorter wavelength than the incident light [43].

Instead of using a single, focused spot of light, a patterned illumination can be used. In this case, a set of patterns must be used, and the resulting signals must be demodulated to produce an image.

### 2.3. Combining the Geometries

### 2.3.1. Confocal Microscopy

Historically, the first approach to combine conventional and scanning imaging techniques was confocal microscopy. The principle of confocal microscopy is illumination by a spot of light and detection of the light coming from a small *confocal* region of the object using a lens focused on a pinhole in front of a detector. Confocal microscopy has two main advantages over conventional microscopy. First, the pinhole rejects light that comes from anywhere other than the confocal spot. Not only does this remove background, stray light, it also filters out light from defocused regions of the object, giving rise to an optical sectioning property. Second, the overall PSF, $H$, for the confocal system is given by the product of the PSFs for the illumination and the detection systems, resulting in, in principle, an improved resolution [5]:

$$H(v) = H_1(v)H_2(v), \tag{2}$$

where $H_1, H_2$ are the PSFs for illumination and detection, and the transverse optical coordinate $v$ is defined in terms of radius $\rho$ as $v = 2\pi\rho \sin\alpha/\lambda$, so one Airy unit 1AU (the distance to the first dark ring of the Airy disk) corresponds to $v \approx 3.83$. The spatial frequency cut-off is equal to the sum of the cut-offs for illumination and detection [5]. The OTF for confocal fluorescence is given by the convolution of the OTFs for the illumination and detection systems, $C(m) = C_1(m) \otimes C_2(m)$, where $m$ is the spatial frequency.

These improvements in resolution can be explained by a trade-off of field-of-view, or after building up a complete image, by trade-off of imaging time.

In 1940, Goldmann described a system where the object was illuminated with a line of light focused from a slit and the image of the illuminated line imaged through a second slit [44]. Thus, this is basically a confocal line illumination system: imaging along the line of illumination is conventional. Actually, Goldmann's system was more clever than that (Figure 1). The principle of Goldmann's microscope is that the axes of illumination (AB) and imaging (CD) are offset by an angle of 45°. Points along the illumination axis are imaged into a line EF at right angles to it. The imaging lens has a magnification of unity so that an aberration-free image can be formed even though the tube length varies as a result of variation in the distance from the plane of the imaging lens. The sample is illuminated by a plane of light using the slit A. A slit is placed to image light within a certain range of the axially illuminated region. Even for a finite slit width, the whole of this region is imaged without aberration. By scanning the object in the direction along AB, a depth image can be integrated. If the width of the detection slit is made very narrow, the system effectively becomes a confocal system. However, it is a confocal system with offset axes, as in the specular microscope [45], theta microscope [46], or dual-axes microscope [47]. These systems all give an enhanced optical sectioning property, as a result of angular gating [48,49]. In the plane perpendicular to the slits, the overall PSF is given by the product of the illumination and detection PSFs. Note that the illumination plane through AB being imaged into the plane through EF is a special case of the Scheimpflug condition, valid for paraxial conditions, which states that the object plane, the image plane, and the median plane of the lens all intersect [50]. In Goldmann's system, the image was detected using a photographic film that was scanned beneath the slit F.

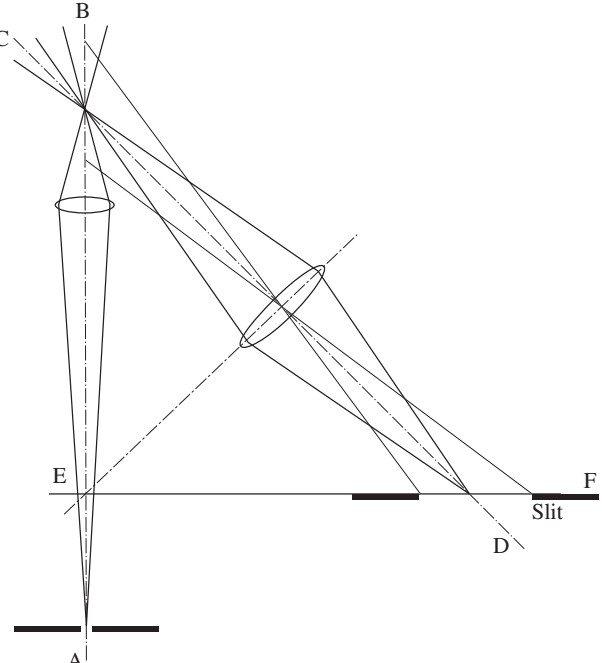

**Figure 1.** Goldmann's optical system. The object is illuminated along the axis AEB. Light is detected along the off-axis path CD.

Koana described, in 1943, a microphotometer system, operating in either a transmission or reflection geometry that used a diaphragm in front of the detector to reject stray light [51]. However, the system did not seem to incorporate scanning, so it did not produce an image as such. Naora carried on with their work and even published a paper in the high-impact journal *Science* [52]. He established the Molecular Biology Unit (later, Research School of Biological Sciences) at the Australian National University in 1968.

A microphotometer system was also described by Zvorykin and Ramberg in 1949 [40], p. 298:

> 'The analysis of photographically recorded spectra is commonly carried out by a microphotometer ... A carefully adjusted, straight tungsten lamp filament is imaged by a microscope objective on the spectrum plate mounted on a mobile stage. A second, similar, microscope objective images the filament image transmitted by the spectrum plate on a slit in front of the cathode of a phototube.'

So this system is in fact a confocal slit system.

Minsky patented the principle of confocal microscopy, which he called a double-focusing optical system, in 1957 [53]. His main motivation was the rejection of scattered light:

> 'This high degree of selectivity afforded by tbe optical system results in a minimum of blurring, increase in signal-to-noise ratio, increase in effective resolution, and the possibility of high resolution light microscopy through unusually thick and highly-scattered specimens.'

He was also aware of the resolution improvement of confocal imaging:

> 'The second pinhole aperture increases the optical resolution of the system by its action of squaring the intensity pattern distribution of the image diffraction. It can be shown that this results in a sharpened central diffraction zone with reduced high order zones.'

He described both transmission and reflection systems.

McCutchen proposed a confocal fluorescence microscope and described how the cut-off frequency was equal to the sum of those for the illumination and detection arms [54]. His impressive paper also discussed near-field microscopy using evanescent waves. In

fact, he regarded the confocal pinhole as forming a synthetic aperture focused on to the object plane.

Petráň and co-workers developed a confocal microscope using arrays of illuminating and confocal pinholes [55,56]. The pinholes were arranged on a spinning disk, thus increasing the scanning speed and allowing the direct observation of a confocal image by eye. Petráň called their system the tandem scanning microscope, as the illumination and detection pinholes were scanned in tandem. Effectively, the spinning disk microscope is like many parallel confocal microscopes. Davidovits and Egger added a laser source to a confocal microscope [57]. To eliminate internal reflections, polarisation filtering and a chopper system were used.

Wied et al. described a computer-controlled cytometer, which had a confocal diaphragm [58]:

'The specimen is imaged onto a measuring diaphragm just in front of the detector photomultiplier by a glycerol immersion quartz objective ... and by a projective lens.'

Slomba described a confocal fluorescence microscope [59]:

'Spurious energy from objects at the same location as the target but which are not in the same focal plane is defocused at [the pinhole] and, therefore, greatly attenuated.'

Their system was also controlled by a computer.

The present author started researching confocal microscopy at the University of Oxford in 1974, under the direction of Professor Rudolph Kompfner. At that time, we knew of no remaining ongoing work on confocal microscopy, although Petráň was still active in Czechoslovakia, behind the iron curtain. The original post-graduate students working on the project were A. Choudhury and J.N. Gannaway. Our first laser scanning microscope was operational in 1975. A theoretical treatment of image formation in confocal microscopes was published in 1977 [5], followed by an investigation of the effect of a finite-sized pinhole [60]. As the pinhole size is increased, the transverse resolution is degraded.

Brakenhoff et al. experimentally demonstrated the improved resolution of confocal transmission microscopy by measuring the PSF [61].

Hamilton et al. experimentally demonstrated the optical sectioning property of a confocal reflection microscope [62]. Cox constructed a confocal microscope with computer control and described the digital processing of confocal images [63]. Cox et al. pointed out that confocal fluorescence microscopy gives super-resolution, as evidenced by the spatial frequency cut-off being given by the sum of the illumination and detection cut-offs [64]. We have explained that for a conventional reflection or transmission microscope, the cut-off is $NA/\lambda$ for on-axis illumination and up to $2NA/\lambda$ for oblique illumination. Similarly, for a conventional fluorescence microscope, the cut-off is also $2NA/\lambda$. Thus, a two-factor improvement is achieved either for oblique illumination or for fluorescence. But, for fluorescence microscopy, no further improvement in cut-off is possible using oblique illumination, as the phase of the wave does not affect the absorption. However, confocal fluorescence microscopy can obtain these two factors of two extensions to the cut-off, resulting in a cut-off of $4NA/\lambda$ if the Stokes shift is neglected.

Confocal interference microscopes were developed by Brakenhoff (transmission mode) and by Hamilton (reflection mode) [65,66].

Illumination using a Bessel beam in a confocal microscope was investigated and patented [5,67,68]. The use of radially polarised light to produce a small, longitudinally polarised spot in the focal region of a confocal microscope was investigated [15,69,70].

A useful theoretical approach to compare the imaging performance of different systems is the 3D OTF [71]. The OTF for a confocal fluorescence system shows that as the pinhole size is increased, the response is degraded [72]. This is a serious problem for confocal fluorescence, as the pinhole must be large enough to obtain a sufficient signal. In practice, a size of 1AU is often used, but this is large enough to remove the resolution ad-

vantage of confocal microscopy. Gan et al. proposed the property of detectability, the ratio of the signal of a point object to the noise from a background volume, as a way to quantify optical sectioning and noise performance [73]. They found that the 3D detectability is optimised for a pinhole size of 0.63AU. This is explained by the argument that too small a pinhole gives insufficient signal, while too large a pinhole reduces optical sectioning, thus introducing noise from defocused regions of the object.

### 2.3.2. Structured Illumination Microscopy

Structured illumination microscopy (SIM) developed in parallel with confocal microscopy. The original proposal was by Lukosz in 1963 [74]:

'A new method is described for obtaining optical images with a resolution exceeding the limits set by diffraction. . . . A mask, or the image of a mask formed by projection is introduced in or near to the object plane. This mask has a variable transmission (for example a grating), and is movable in the object field. A second similar mask is introduced in or near to the image plane, or the plane of an intermediate image, and is moved conjugately with the object plane mask. The image obtained during the scanning by the masks is integrated in time by a receptor of suitable inertia (for example, the eye, or a photographic emulsion). There results an image of the object with enhanced resolution and contrast (the bandwidth of the transmitted spatial frequencies is increased, and the frequency response is raised). The method may be used with coherent, partially coherent or incoherent illumination.'

Lukosz thus proposes using two masks: one to produce the patterned illumination and another to demodulate the signal. Therefore, the reconstruction is performed optically, as compared with the usual current way of digital processing.

Note that the achievable cut-off frequency is the same as for oblique illumination, but the bandwidth is twice as large. Hence, if one uses oblique illumination in a coherent interference microscope to obtain phase information, using two angles of incidence would give a similar result to SIM by a process of synthetic aperture imaging.

Neil et al. used digital processing to give an optical sectioning effect in fluorescence microscopy [75]. They produced a fringe pattern with two interfering beams and used a simple nonlinear reconstruction algorithm similar to a phase-shifting algorithm. They found that the optical sectioning is maximised when the NA of the illuminating pattern is one half of the NA of the imaging lens.

Hanley et al. constructed a programmable array microscope using programmable illumination of excitation/detection patterns, such as Hadamard masks [76].

Heintzmann and Cremer used a finer fringe projection to give almost a doubling in the spatial frequency bandwidth [77]. The fringes were generated using a diffraction grating imaged on to the object. Their reconstruction algorithm separated the spatial frequencies from each of the fringe grating orders and relocated (downshifted) them in 3D Fourier space. The fringes must be rotated relative to the sample in order to produce a nearly isotropic image. Therefore, there are two different versions of 2D SIM, according to whether the aim is maximum transverse resolution or 3D imaging. Gustafsson achieved an improvement in spatial resolution of a factor of two by recombining the components using weighted averaging in the Fourier space [78]. Gustafsson et al. went on to obtain a doubling in resolution in all 3Ds [79]. This approach, called 3D SIM, uses patterned illumination from a zero and two diffracted grating orders.

Two specific features are important in achieving such a high-resolution performance. First, the object is illuminated with two plane waves at close to the angle $\alpha$ of the objective lens. Then the illuminating intensity along the $x$ direction is $I(v_x) = \cos^2 v_x$, which is narrower than that produced by a slit pupil, $I(v_x) = (\sin v_x / v_x)^2$, and even more narrower than that produced by a circular aperture, as in confocal microscopy, $I(v_x) = (J_1(v_x)/v_x)^2$, where $J_n$ is a Bessel function of order $n$. For a Bessel beam, $I(v_x) = J_0(v_x)^2$. The different values of $v_x$ for $I$ drop to one half for these four cases: 0.79, 1.39, 1.62, and 1.12, respectively.



Thus, the SIM case gives a substantial improvement over any of the others, enhancing the higher spatial frequencies. Second, the interfering illuminating beams are arranged to be always *s*-polarised to maximise fringe contrast. This is equivalent, in a confocal microscope, to using azimuthal polarisation with a phase vortex [70,80,81].

### 2.3.3. Digital Deconvolution

Confocal microscopy was first commercialised in 1982 by a start-up company, Oxford Optoelectronics, Ltd. [82]. The technique became popular in the late 1980s and early 1990s, and several companies manufactured systems. A commercial competitor in those days was digital deconvolution of conventional microscope images. The key was to process a stack of defocused images, which then provided the sufficient information content for deconvolution. The earliest example of this approach was by Castelman and coworkers [83], followed by Agard and Sedat [84]. The deconvolution process required the use of a theoretical or measured PSF. Later, blind reconvolution algorithms were developed [85]. Several companies marketed programs. Interestingly, one of the companies described their system as performing *photon reassignment*. Eventually, confocal microscopy became the method of choice, but deconvolution remained widely used to process confocal images, where it could reduce apparent image noise. Developments are still continuing, with the incorporation of total variation regularisation, sparsity, deep learning, and the use of graphics cards [86–89].

### 2.3.4. Confocal Microscopy with a Detector Array

We have already mentioned that the spinning disk confocal microscope has an array of detector pinholes [56], whereas a point-scanning confocal microscope uses just a single detector pinhole. Many papers have discussed replacing the pinhole of the confocal microscope with a detector array. Bertero and Pike proposed using a detector array to give multiple images that were then digitally deconvolved [90]. Reinholz et al. used a detector array to measure the spread light distribution while scanning [91]. They observed shifts of the main maxima and sensitive behaviour of the side lobes in the image and claimed that these could be used to extract more information. They called tracking the maximum in the detector plane a Type III microscope, in contrast to a Type I microscope (nonconfocal scanning) or Type II microscope (confocal microscope). This approach could be useful in confocal transmission microscopy, where the confocal spot may be deflected by refractive index inhomogeneities in the object. Sheppard and Gu investigated the image of an edge formed by the maximum signal [92]. They calculated the image of the edge in the detector plane, showing that the PSF is distorted. They found the intensity at the edge itself for confocal microscopes with a finite-sized pinhole or slit. Barth and Stelzer investigated using a detector array to enhance resolution [93]. A detector array can also be used to align the confocal pinhole or to adjust its size post-imaging [94,95].

Benedetti described a system using multi-point illumination and a detector array [96]. The image from the detector consists of sub-images from each illumination spot. He called their system video confocal microscopy [97]. Different images can be reconstructed by averaging or determining the maximum or minimum value in the sub-image. He also proposed super-confocal imaging, defined as $max + min - 2avg$, which exhibits exceptional optical sectioning characteristics. More recently, he has applied statistical processing of the sub-images, based on moments, giving impressive results [98].

Chen et al. proposed the technique of focal modulation microscopy, where the illuminating spot is modulated spatiotemporally, and the signal, after passing through a pinhole, is detected by a lock-in amplifier [99]. The illumination consists of fringes modulating an Airy disk, so it resembles a scanning version of SIM, where the phase of the fringes continuously varies in time. The aim was originally to improve penetration of a scattering object, but it was found that the transverse resolution was also improved [100]. Lu et al. proposed a system with spatiotemporal modulation [101]. One scheme was called scanning patterned illumination (SPIN) microscopy and employs the modulation of

the excitation combined with temporally cumulative imaging by a nondescanned array detector. The second scheme, scanning patterned detection (SPADE) microscopy, used the modulation of the detected signal together with spatially cumulative imaging by a nondescanned single-element detector. Lu et al. proposed spatiotemporal modulation by mathematical processing of digital images, which they called virtually structured detection (VSD) [102]. Laporte et al. explored resolution enhancement in nonlinear scanning microscopy by saturated absorption or stimulated emission through post-detection digital computation [103].

Many papers have explored the technique of subtractive microscopy, where the signals from two different pinhole sizes are subtracted to give an improvement in resolution [104–108]. Usually, subtraction is detrimental, however, because it tends to decrease the signal while also increasing the noise. This problem can be avoided by using a ring-shaped detector array rather than subtracting [104]. Further, the OTF for confocal fluorescence with a finite-sized pinhole exhibits negative regions [72], so subtraction actually increases the signal [105]. Heintzmann et al. demonstrated subtractive imaging using weighted-averaging in Fourier space [106]. Sanchez-Ortiga et al. used a charge-coupled device (CCD) detector [107]. Korobchevskya et al. used a computational approach to determine the optimum subtraction coefficient [108]. The technique of focal modulation microscopy is also closely related to subtractive microscopy, except that the subtraction is performed by temporal modulation, which decreases the bandwidth and hence the detector noise [99].

### 2.3.5. Structured Illumination Microscopy versus Confocal Microscopy

A comparison of the image formation properties in confocal microscopy and SIM was given in Ref. [109]. Optically, confocal microscopy uses a large number of illumination patterns with a small number of detection elements, whereas SIM uses a small number of illumination patterns and a large number of detection pixels. It should be appreciated that there are also some similarities between the two modes. For example, in the spinning disk confocal microscope, the object is illuminated with a pattern of spots, which can be considered, by considering its Fourier series, a 2D fringe pattern. The main difference of the spinning disk microscope from SIM is in the mode of reconstruction. In the spinning disk microscope, the reconstruction is achieved optically using the detection pinholes, in a similar fashion to Lukosz's original SIM [74]. However, in modern SIM, the images are processed digitally, usually in Fourier space.

### 2.3.6. Image Scanning Microscopy

We have already discussed the use of a detector array in a confocal microscope. The name image scanning microscopy (ISM) was introduced by Mueller and Enderlein to describe their microscope system using a detector array [110]. It is not clear exactly to which feature they meant for this term to refer to. The main novel feature is obviously not the addition of a detector array to the confocal microscope, as this was not a new idea. There are two concepts that they describe. First, the recognition that an off-axis element of the detector does not represent a signal from the point of the sample that is being illuminated. This means that processing of the data must take account of this effect. Second, the most simple way to process the data is to shift the measured signal to the coordinates of the object to which it refers. This operation is called pixel reassignment. Thus, our understanding of the terms is that pixel reassignment is a particular strategy of processing the data, whereas ISM can refer to methods of processing other than simple pixel reassignment.

In fact, both the principle of image scanning microscopy and pixel reassignment were introduced much earlier [111]. It was mentioned that such a system is especially useful for confocal fluorescence, as the resulting signal is much stronger than in a confocal microscope with a small pinhole. The case of a confocal reflection system was also discussed. This paper also gave the PSF and OTF for a system with an infinitely large detector array after summing the reassigned pixel signals. The overall PSF is the convolution of the two rescaled

(compressed) PSFs, $h(v) = h_1(2v) \otimes h_2(2v)$, while the OTF is the product of two rescaled (enlarged) OTFs, $C(m) = C_1(m/2)C_2(m/2)$. Note that for confocal microscopy with a small pinhole, the overall PSF is given by a product and the OTF by a convolution. A plot of the OTF was given, showing that the high spatial frequency response is increased even compared with confocal imaging with a small pinhole. The reason for this is that, as was already known, a displaced point detector, surprisingly, results in a sharper PSF [112–114]. It is interesting to note that these papers assumed that the image had been reassigned, but as only a single displaced detector was used, no reassignment was necessary. Thus, ISM gives a much stronger signal while, at the same time, giving better resolution than true confocal microscopy, with an infinitely small pinhole.

It was also explained that the image formation model justifies why resolution is improved in a confocal microscope with a small pinhole, whereas you might think that throwing away light would not do this: increasing the size of the pinhole introduces light coming from neighbouring parts of the object, thus reducing the resolution. Thus, the confocal pinhole actually adds information about the position in the detector plane. A detector array can therefore be seen to add even more information.

Mueller and Enderlein were the first to perform ISM experimentally [110]. They used an EMCCD detector, applied pixel reassignment and summation, and then also used post-processing by linear deconvolution to obtain a doubling in resolution.

There are many possible alternative ways to process the ISM data. The measured signal from a 2D object is a function of four variables: $S(\mathbf{x}_1, \mathbf{x}_2)$, where $\mathbf{x}_1, \mathbf{x}_2$ are coordinates for the scan position and the detector location. The 4D signal in a reflection-mode spinning disk microscope was presented many years ago and applies equally to ISM [115]. Other papers that consider imaging in spinning disk microscopes include Refs. [116,117]. The 4D nature of the signal was also apparent in Refs. [91,96]. For fluorescence ISM, the 4D signal is [118]

$$I(\mathbf{x}_1, \mathbf{x}_2) = \iint H_1(\mathbf{x}_1 - \mathbf{x}_0) H_2(\mathbf{x}_2 - \mathbf{x}_0) t(\mathbf{x}_0) d^2\mathbf{x}_0, \tag{3}$$

where $\mathbf{x}_0$ is the coordinate vector in the object plane, which is basically a cross-correlation function. We can see that $\mathbf{x}_1 = \mathbf{x}_0$ gives a conventional image, $\mathbf{x}_2 = \mathbf{x}_0$ gives a scanning image, and $\mathbf{x}_1 = \mathbf{x}_2$ gives a confocal image. If $H_1 = H_2$, the signal for $\mathbf{x}_1, \mathbf{x}_2$ gives an image of the point $(\mathbf{x}_1 + \mathbf{x}_2)/2$, i.e., midway between the illumination point and the detected point. Importantly, it does not give an image of the illuminated point $\mathbf{x}_1$. Of course, the PSFs $H_1, H_2$ are compact, so the complete 4D signal contains redundant information.

Soon after my paper was published, Bertero et al. published a paper where they applied, almost as an afterthought, deconvolution by Tikhonov regularisation to 1D pixel reassignment[119]. Their conclusion was

'It is seen that Sheppard's method provides an improvement in resolution with respect to conventional CSLM which is smaller than the improvement provided by the method discussed in this paper.'

However, in my opinion, this was not a fair comparison, as better results can be obtained by multi-image deconvolution of ISM data. Further, pixel reassignment is much simpler than their algorithm. Later, however, they further investigated a more general algorithm, for which the Bertero et al. algorithm and the pixel reassignment algorithm were special cases [120].

A disadvantage of SIM is that thick samples distort the fringes and hence limit the sample thickness. York et al. described a system that could image samples eight-times thicker than SIM while retaining a comparable resolution and high speed (one 2D image/s) [121]. Their name 'multifocal SIM' can be explained in terms of their view of their system as SIM with multi-point illumination, but we prefer to consider it as a multifocal ISM. In fact, they used exactly the pixel reassignment and summation algorithm of ISM. Multi-focal ISM is possible because only a small region of pixels gives an appreciable signal for each illumination spot. In a later paper, they increased the speed to $100 \times$ 2D image/s by performing the reconstruction optically, using a matched microlens array to locally

contract each pinhole emission, with a galvanometric mirror to translate the excitation pattern and the summation of the fluorescence emissions with a camera [122].

De Luca et al. used a galvonometric mirror to perform the pixel reassignment and scaling [123]. The reassignment factor can be altered as required. They called their method Rescan Confocal Microscopy. The system is similar to an architecture for confocal microscopy with a pinhole and without pixel reassignment proposed by Brakenhoff and Visscher [124].

Schultz et al. constructed a confocal spinning disk ISM [125]. A CCD detector images a region of pixels around the centre of each illumination spot. A similar system was described by Azuma and Kei [126].

Roth et al. described an ISM with optical pixel reassignment [127]. They called their system OPRA (Optical Photon Reassignment Microscopy). Later, they combined OPRA with patterned illumination to improve axial imaging [128].

Sheppard et al. presented a theoretical analysis of ISM [129]. This paper introduced the term pixel reassignment. ISM with an infinitely large detector array gives no optical sectioning. In fact, the size of the detector array plays a similar role to the size of the pinhole in confocal microscopy. It was shown how the resolution and signal collection efficiency of ISM varied with the size of the detector array. The FWHM of the PSF for ISM was found to be about 10% narrower than for confocal microscopy with a small pinhole. The pixel reassignment strategy was shown to be valid for any value of the pixel reassignment factor, with an optimum being to reassign to the point midway between the illumination and detection points for a system with no Stokes shift. In the presence of a Stokes shift, the reassignment factor should be altered. It was pointed out that although optical methods of reassignment could be used, there was some benefit in recording the complete data in order to retrospectively choose the pinhole size and for subsequent multi-image processing.

Castello et al. recognised that only a small number of detector elements in ISM measure a significant signal and proposed using a quadrant avalanche photodiode detector array, i.e., just four pixels [130]. Even with this small number of elements, this is enough to attain a resolution close to an ideal confocal microscope and increases the signal-to-noise ratio by a factor of 1.5.

Ingaramo and Winter et al. applied the multifocal ISM approach to two-photon excitation and used multi-image deconvolution [131–133].

Zeiss introduced the Airyscan microscope, which uses a hexagonally arranged fibre bundle to couple to photomultiplier detectors, simulating a detector array [134]. Korobchevskaya et al. explored the imaging performance of Airyscan [135]. Sivaguru et al. compared the performance of Airyscan and SIM [136]. They found that SIM gave a better resolution for thin samples, but Airyscan was preferable for thicker samples. Scipioni et al. investigated the use of the Airyscan for fluctuation correlation spectroscopy [137].

As the PSF in ISM is narrower than in a conventional microscope while almost all the light is detected, Roth et al. investigated the property that the peak of the PSF is higher than physically achievable in an ordinary optical system [138]. They called this effect superconcentration. In fact, this behaviour was also evident in the plots of Ref. [129].

Sheppard et al. pointed out that for a comparison of OTFs from different optical systems, the absolute strength of the signal is an important factor [139]. Hence, for ISM, the absolute value of the spatial frequency response is much higher than in confocal microscopy with a small pinhole, as most of the light is detected. The unnormalised OTF for confocal microscopy and ISM with different sizes of pinholes/arrays were presented. Although for confocal imaging, the OTF can exhibit negative values for pinholes larger than 0.5 AU, this does not occur for ISM.

Gregor et al. described a two-photon excitation ISM based on a form of the rescan technique [140].

Sheppard et al. analysed the imaging properties of ISM in terms of the 4D signal, $S(\mathbf{x}_1, \mathbf{x}_2)$, and proposed reconstruction of the 4D signal in 4D Fourier space [118]. The imaging properties of 2PF ISM were investigated. The pixel reassignment factor can be

chosen to optimise the OTF for a given noise level. The effect of the reassignment factor on image formation in a single photon fluorescence ISM with a Stokes shift was also studied. The effects of using Bessel beam illumination or pupil filters on the OTF for ISM with a large detector array were investigated.

Castello et al. developed an ISM system using a novel 25 element ($5 \times 5$) single-pixel avalanche photodiode array detector (SPAD) and demonstrated its application for FLIM (fluorescence lifetime microscopy) [141]. Pixel reassignment was achieved by an automatic process based on cross-correlation between images from different detector pixels. Multi-image deconvolutions using a Richardson–Lucy algorithm or Wiener filtering were presented. They demonstrated the ability to achieve the same resolution as ideal confocal microscopy at one-tenth of the illumination intensity.

Roider et al. reported applying ISM to Raman microscopy [142]. Roider and Tzang et al. described ISM in single-photon and 2PF fluorescence modes using PSF-engineering to obtain a 3D image in a single 2D scan [143,144].

Breedijk et al. proposed using annular illumination to improve the transverse resolution [145].

Koho et al. described 2PF ISM with a SPAD detector and blind deconvolution [146].

The group of Kuang and Liu has published many relevant papers [147–151]. They proposed a subtractive imaging method with pixel reassigned and non-reassigned images [150], also using doughnut illumination [147]. They explored ISM with maximum-likelihood estimation [148]. They investigated reassignment based on image cross-correlation [149]. They investigated 2PF ISM [151].

Ye and McCluskey investigated pixel reassignment in a reflection microscope [152]. Mozaffari et al. used pixel reassignment in an ophthalmoscope [153]. Strasse et al. reported a spectral ISM [154]. Mandracchia et al. used pixel reassignment in an optofluidic microscope [155]. Wang et al. used pixel reassignment in a SHG microscope [156]. Dan et al. used pixel reassignment to reconstruct SIM images [157]. McGregor et al. investigated different post-processing strategies in ISM [158].

Sheppard et al. investigated the effect of altering the pixel reassignment factor as a function of the detector pixel offset and proposed reassigning it to the peak of the PSF for a particular detector element [159]. They included array sizes up to 2 AU, for which the peak of the PSF is 1.9 times that of a conventional microscope. They found that although the PSF from a single offset detector pixel can become distorted, after its integration around a ring of elements, the PSF is compact and well-behaved. The axial resolution was found to be higher than that for a confocal microscope with a pinhole of the same size as the array. It was found that the detectability for ISM is optimised for an array of 1.1 AU. The effect of weighting the array signals was also considered. This study was extended to the cases of Bessel beam illumination and to 2PF and 3PF [160]. Although Bessel beam illumination gives poor imaging in ISM with a large array, the PSF is well-behaved for a small array. A pixel offset of 0.836 AU corresponds to the case when the PSF for that pixel exhibits two peaks of equal height.

The PSF from a detector pixel can be interpreted as showing the probability density function of the signal, the peak giving the most likely origin in the object of a detected photon. It has been pointed out that reassigning the maximum of the PSF for each detector pixel is similar to a maximum likelihood restoration [161]. Pixel reassignment was applied to the case of a vortex doughnut beam. The transverse width of the PSF was found to be smaller than in a conventional imaging system for any size of detector array and smaller than a true confocal microscope (point detector) for arrays larger than about 1.3 AU. The axial resolution was greater than in a conventional imaging system for any size of array and greater than in a confocal microscope for arrays larger than about 0.9 AU. The optical sectioning was stronger, even than for ISM with Airy disk illumination, for array sizes larger than 1.1 AU.

Yu et al. proposed that in the image in the detector plane, sparsity is imposed by the illumination spot, which greatly improves the deconvolution of this data [162]. They

claimed resolving 70 nm resolution test objects. Subsequently, they used $l_1$ regularisation and claimed a resolution of 60 nm [163].

Fluorescence is basically an incoherent effect, as there is no phase correlation between fluorescent molecules. Weak fluorescence displays quantum effects, such as antibunching. Fluorescence also exhibits blinking, which has been exploited in super-resolution optical fluctuation imaging (SOFI), where low order or higher order time correlations are used to extend the spatial frequency cut-off [164,165]. The group of Oron has exploited both antibunching (Q-ISM: Quantum ISM) and time correlations (SOFISM: SOFI-ISM) in ISM [166,167]. The signal measured in Q-ISM depends on the square of the excitation PSF but also on the second power of the detection PSF; the nonlinearity in detection translates into a possibility for up to a four-fold improvement over the diffraction limit. The combination of correlation in temporal fluctuations and ISM yields a $\times 4$ resolution enhancement for the second-order correlation.

Reviews of ISM have been presented by Chen et al. and Ward and Pal [168,169]. A comparison of the imaging properties of ISM and SIM was given in Ref. [109]. The resolution of different systems, normalised to that of a conventional microscope, are shown in Table 1. These figures assume optimum reassignment and summation for ISM, and no processing except shifting of spatial frequencies for SIM.

**Table 1.** Resolution improvement for different systems.

| System | Resolution Improvement Factor |
|---|---|
| Conventional | 1 |
| 2-beam (a) SIM | 1.40 |
| 2-beam (b) SIM | 2.26 |
| 3-beam SIM | 1.71 |
| Confocal, ideal | 1.39 |
| Confocal, ideal, with Bessel beam | 1.72 |
| ISM, 2 AU array | 1.53 |
| ISM 0.836 AU array, with Bessel beam | 1.82 |

## 3. Discussion

The techniques of confocal microscopy and image scanning microscopy (ISM) have been reviewed. Confocal microscopy is an established technique, which is widely used both in biomedical imaging and in industrial and material examinations. ISM is a comparatively new technique. When proposed, suitable detectors for its implementation were unavailable, so it took two decades before it was experimentally demonstrated. Basically, it can be performed simply by replacing the pinhole and detector of a confocal microscope with a detector array. It seems likely that, in many cases, it will replace the existing form of the confocal microscope. The most straightforward way to process the image is using pixel reassignment, where the signal from a detector element is assigned to the most likely origin in the sample. There are two main ways this can be implemented: by optically reassigning the data and recording a reconstructed image directly or by storing all the image data and digitally processing. Both approaches have their merits: the first is faster, but the second allows more flexibility and more advanced digital processing. For the second approach, as a detector array with only a small number of elements is required, multiplexing by multi-point illumination is also available. We expect further improvements in image reconstruction algorithms in the future based on developments in deconvolution on the one hand and using the statistical properties of the imaging data on the other.

**Funding:** This research received no external funding.

**Institutional Review Board Statement:** Not applicable.

**Informed Consent Statement:** Not applicable.

**Data Availability Statement:** Not applicable.

**Acknowledgments:** They author acknowledges the support and useful discussions with the staff of Istituto Italiano di Tecnologia, Genova, Italy, especially Marco Castello, Giorgio Tortarolo, Giuseppe Vicidomini, and Alberto Diaspro.

**Conflicts of Interest:** The author declares no conflict of interest.

## Abbreviations

The following abbreviations are used in this manuscript:

| | |
|---|---|
| CCD | Charge Coupled Device |
| CTF | Coherent Transfer Function |
| FMM | Focal modulation microscopy |
| FWHM | Full-Width at Half-Maximum |
| ISM | Image Scanning Microscopy |
| OPRA | Optical Photon Reassignment Microscopy |
| OTF | Optical Transfer Function |
| PSF | Point Spread Function |
| Q-ISM | Quantum Image Scanning Microscopy |
| SHG | Second Harmonic Generation |
| SIM | Structured Illumination Microscopy |
| SNR | Signal-to-Noise Ratio |
| SOFI | Super-resolution Optical Fluctuation Imaging |
| SOFISM | Super-resolution Optical Fluctuation Image Scanning Microscopy |
| SPADE | Scanning Patterned Detection |
| SPIN | Scanning Patterned Illumination |
| TCC | Transmission Cross-Coefficient |
| THG | Third Harmonic Generation |
| 2D | Two-Dimensional |
| 2PF | Two-Photon Fluorescence |
| 3D | Three-Dimensional |
| 3PF | Three-Photon Fluorescence |
| 4D | Four-dimensional |

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
