# Peer review of "The Development of Microscopy for Super-Resolution: Confocal Microscopy, and Image Scanning Microscopy"

_applsci, doi:10.3390/app11198981_

Round 1
Reviewer 1 Report
Recommendation: Accept after minor revision
Comments:
The manuscript entitled “Superresolution: Confocal microscopy, and image scanning microscopy.” reviews the history of the development of super-resolution microscopes. The author provided a very detailed but big picture to this field, which also benefit me. The manuscript is in a reasonable order: firstly, the literature reviewed the super-resolution and then the three types of microscopes (non-confocal, confocal, and image scanning microscopes). The comparisons, advantages, and disadvantages for each of the techniques are provided, which is solid and detailed. However, the absence of the near field scanning optical microscope (NSOM or SNOM) is a pity in this manuscript, which is entitled “super-resolution” especially. As far as I know, there are two types of NSOM or SNOM, namely the scattering (apertureless) type, and the aperture type. The former one has higher resolution but limited coupling efficiency which leads to a lower signal-to-noise ratio whereas the latter one has the trade-off between the high transmittance efficiency and high resolution (aperture size). Currently, the scattering-type NSOM can use the interferometer to perform the higher-order harmonic signal to get rid of its low signal-to-noise ratio where the aperture type NSOM can use the advanced nano-focusing methods to the nano-focus hot spot at a solid apex of the probe. The nano-focusing methods include but are not limited to the remote coupling (Nano letters 7, no. 9 (2007): 2784-2788.; Nano letters 19, no. 1 (2018): 100-107.) the tapered optical fiber to plasmonic structure coupling (Nature Photonics 13, no. 9 (2019): 636-643.) or the insulator–metal-insulator configurations (Nature Photonics 8, no. 1 (2014): 13-22.).
Besides the issues illustrated above, here are some minor problems:
- In the abstract section, the author claimed: “The main advantage of optical microscopy, over other types of microscopy such as electron microscopy, is its noninvasive nature.”. I can not agree that the electron microscopies include SEM and TEM are noninvasive. The high-energy electron beam can damage the surface of most of the samples. Take an extreme example, the photoresist sample can be exposed under the electron beams. (It is the actual way how the E-Beam lithography works).
- Please check the typo in the manuscript, for example, in line 219, the word should be “objective” rather than “ohjective”.
- Please check the format from page 6 to page 8.
Overall, this is a very excellent review paper. If the authors can solve these problems, I would like to recommend this paper can be published in Applied Physics after a minor revision.
Reviewer 2 Report
The reviey by Sheppard is very nice and comprehensive. I enjoyed reading it and getting a good overview over the development and the history of (confocal) microscopy. I can therefore recommend publication basically as is (except for some typos which I found: see below).
However, I suggest to change is the title and the abstract since from them the reader expects a topic to be discussed which is not addressed sufficiently. I think the review is about the development of microscopy toward superresolution and not about what is normally called "superresolution". Also the first sentence in the abstract "Superresolution microscopy is reviewed." is misleading. If this was the case, the principles of STED and localization-based methods would have to be discussed in detail.
l. 79: transmission
formula 1: define s and n
l. 98: signal
l. 242: called
l. 298: verb "was" is missing
l. 327: The
l. 411: detection
